# A Multifaceted Intervention and Its Effects on Antibiotic Usage in Norwegian Nursing Homes

**DOI:** 10.3390/antibiotics12091372

**Published:** 2023-08-27

**Authors:** Nicolay Jonassen Harbin, Jon Birger Haug, Morten Lindbæk, Per Espen Akselsen, Maria Romøren

**Affiliations:** 1Antibiotic Center for Primary Care, Department of General Practice, Institute of Health and Society, University of Oslo, 0450 Oslo, Norway; morten.lindbak@medisin.uio.no; 2Department of General Practice Institute of Health and Society, Faculty of Medicine, University of Oslo, 0450 Oslo, Norway; maria.romoren@medisin.uio.no; 3Department of Infection Control, Østfold Health Trust, Kalnes, 1714 Grålum, Norway; jon.birger.haug@so-hf.no; 4Norwegian Centre for Antibiotic Use in Hospitals, Department of Research and Development, Haukeland University Hospital, 5021 Bergen, Norway; per.espen.akselsen@helse-bergen.no

**Keywords:** antibiotic stewardship, academic detailing, quality improvement, nursing homes, urinary tract infection

## Abstract

We explored the impact of an antibiotic quality improvement intervention across 33 nursing homes (NHs) in one Norwegian county, compared against four control counties. This 12-month multifaceted intervention consisted of three physical conferences, including educational sessions, workshops, antibiotic feedback reports, and academic detailing sessions. We provided clinical guiding checklists to participating NHs. Pharmacy sales data served as a measure of systemic antibiotic use. The primary outcome was a change in antibiotic use in DDD/100 BD from the baseline through the intervention, assessed using linear mixed models to identify changes in antibiotic use. Total antibiotic use decreased by 15.8%, from 8.68 to 7.31 DDD/100BD (model-based estimated change (MBEC): −1.37, 95% CI: −2.35 to −0.41) in the intervention group, albeit not a significantly greater reduction than in the control counties (model-based estimated difference in change (MBEDC): −0.75, 95% CI: −1.91 to 0.41). Oral antibiotic usage for urinary tract infections (UTI-AB) decreased 32.8%, from 4.08 to 2.74 DDD/100BD (MBEC: −1.34, 95% CI: −1.85 to −0.84), a significantly greater reduction than in the control counties (MBEDC: −0.9, 95% CI: −1.28 to −0.31). The multifaceted intervention may reduce UTI-AB use in NHs, whereas adjustments in the implementation strategy may be needed to reduce total antibiotic use.

## 1. Introduction

In Norway, nursing homes (NHs) account for 6–7% of human antibiotic use, while hospitals contribute 8% [1]. Prior research has indicated a considerable variation in antibiotic use between different NHs, and even in homes with similar resident compositions, with no clear rationale behind these variations [2,3,4]. While a benchmark for antibiotic use has not yet been established, the divergence suggests potential room for improvement in antibiotic prescription practices within NHs.

Residents of NHs often have frail health, comorbidities, and immunosenescence, rendering them more prone to infections [5]. Diagnosing and treating infections in this demographic are challenging medically and ethically due to cognitive impairment, ambiguous clinical presentation of symptoms, and limited diagnostic resources [5,6,7]. Factors such as polypharmacy, influence from next-of-kin, and part-time physician coverage further complicate the diagnosis and treatment of infections [6,8,9].

The most common indication for antibiotic use in NHs is urinary tract infections (UTIs), including prophylaxis, usually followed by respiratory tract infections (RTIs) [4,10,11,12,13,14]. However, due to the high prevalence of asymptomatic bacteriuria (ABU), in combination with uncertain and challenging diagnosis in the NH population [15,16], many courses of antibiotics prescribed for suspected UTIs are likely unnecessary. Indeed, a significant proportion of antibiotic prescriptions targeting suspected UTIs in NHs have been deemed inappropriate [17,18,19], making UTIs a clear target for interventions designed to reduce and improve antibiotic use in NHs.

In response to the escalating threat of antimicrobial resistance (AMR), the World Health Organization (WHO), the European Union, and several countries have released action plans in the last decade [20,21,22]. The Norwegian Government unveiled its “National Action Plan against Antibiotic Resistance in the Health Service” in 2016, aiming to reduce human antibiotic consumption by 30% by 2020 relative to the 2012 consumption level [23]. This plan included specific measures targeted at NHs.

Most Norwegian NHs employ general practitioners (GPs) who split their time between the NH and their own practices. While some large NHs have full-time NH physicians, many do not offer daily physician coverage. Therefore, NH nurses are essential in observing and diagnosing acute-onset illness in NH residents [6]. In situations of suspected infections in residents, NH nurses typically contact the permanent NH physician or an out-of-hours physician if the NH physician is unavailable [24]. Physicians contacted by phone may diagnose an infection and prescribe antibiotic therapy without a clinical examination based on the nurses’ observations, point-of-care investigations such as urinary dipstick and C-reactive protein analysis, or a resident’s previous infection history. In cases where residents are hospitalised, the hospital physicians may initiate antibiotic treatment during the hospital stay, which is then continued for some time after discharge back to the NH. The majority of nursing homes (NHs) maintain an inventory of the most frequently utilised antibiotics, ensuring immediate access when required. In instances where a specific antibiotic is not readily available, orders are placed with the pharmacy to replenish the stock. Since 2008, the National Guidelines for Antibiotic Use in Primary Care have incorporated a distinct section dedicated to NHs. These guidelines are widely recognised and highly regarded among Norwegian NH and primary care physicians, serving as a key resource in guiding antibiotic use within these settings.

To alleviate growing pressure on the hospital system, the Norwegian Care Coordination Reform was introduced in 2012 [25], allocating more resources to the primary healthcare sector and allowing intravenous (IV) medication, including antibiotics, to be initiated and administered in NHs [24]. However, the use of IV antibiotics allows more use of broad-spectrum antibiotics, which can contribute to antibiotic resistance. A recent study found that IV antibiotics made up 17% of the total antibiotic purchases from pharmacies by Norwegian NHs in 2016 [4], a significantly higher proportion than found in a similar 2007 study where IV-ABs constituted 1% of the total antibiotic purchase [2].

Over the past two decades, several international multifaceted antibiotic stewardship programs (ASPs) in NHs have demonstrated promising results in improving antibiotic use [26,27], with academic detailing, in particular, proving successful in both general practice and NH settings [28,29]. In line with the national action plan, the Antibiotic Centre for Primary Health Care designed the “RASK” intervention (Riktigere Antibiotikabruk for Sykehjem i Kommunene; Prudent Antibiotic use for Municipal Nursing Homes), a nationwide, multifaceted quality improvement program targeting all levels of healthcare staff in NHs. Enrolment of NHs was performed on a county-by-county basis, with approximately four counties each year, until all Norwegian counties were included.

The aim of the study was to present the effectiveness of this government-backed antibiotic quality improvement intervention in NHs in the county of Østfold, with four other counties serving as controls. We assessed the changes in (1) total antibiotic use, (2) antibiotic use per indication group, (3) guideline-compliant antibiotic use, and (4) methenamine use within the intervention and compared them with the changes in the combined control counties.

## 2. Results

From 12 October 2016 to 11 October 2017, 33 of the 37 (89%) nursing homes (NHs) in Østfold County completed the intervention. Two NHs did not respond to the invitation, and one was undergoing renovation during data collection. One NH was excluded midway through the intervention due to inaccurate data collection. Of the remaining NHs, fourteen were categorised as long-term, two as short-term, and seventeen as mixed (long-term and short-term). Nine were small, fifteen medium-sized, and nine were large NHs. The four control counties—Buskerud (Control 1), Agder (Control 2), Telemark (Control 3), and Vestfold (Control 4)—comprised 35, 42, 25, and 30 NHs, respectively. These control counties did not differ substantially from the intervention county in NH categories but had a higher amount of small-sized NHs and fewer total NH beds compared to the intervention county (Table 1).

### 2.1. Measures and Implementation of Intervention Material

We received the report describing measures taken and implementation of intervention material halfway into the intervention period from 29 of 33 (87.9%) participating NHs. These 29 NHs reported providing their staff with educational lecture material and antibiotic feedback reports. However, the proportion of staff members receiving these educational presentations varied substantially, ranging from 15% to 95% (mean of 52%). The urinary dipstick checklist was regarded as a valuable tool by 28 of 29 NHs (96.5%), 21 of 29 NHs (72.4%) anticipated that the checklist could reduce urine dipstick testing at their institution, and 19 out of 29 NHs (65.5%) reported using the urinary dipstick checklist. Eight NHs (27.6%) reported using the acute-onset illness checklist.

### 2.2. Antibiotic Use (Excluding Methenamine)

The changes in antibiotic use from baseline recorded to the intervention’s conclusion within the intervention county are detailed in Figure 1. These data illustrate a substantial variation in prescribing practices among NHs and within NH subgroups. Changes in antibiotic use totally and within the NH subgroups for both the intervention and control counties are illustrated in Appendix A. For the intervention county and the control counties combined, total antibiotic use was lowest for long-term NHs and highest for short-term NHs. Similarly, medium and large-sized NHs had lower total antibiotic use than small NHs (Table 2).

During the intervention year, the mean overall antibiotic use in the intervention county decreased by 15.8%, from 8.68 DDD/100BDs to 7.31 DDD/100BDs, with a model-based estimated change (MBEC) of −1.38 DD/100BDs (95% CI −2.35–−0.41) (Table 3). Short-term NHs saw a 23.1% decrease, mixed NHs a 14.1% decrease, and long-term NHs an 8.5% decrease. Among the control counties, two demonstrated no statistically significant change in overall antibiotic use (Appendix A). One control county (Telemark) showed a borderline significant reduction (*p* = 0.057), while another (Vestfold) evidenced a significant reduction, with an MBEC of −1.84 DDD/100BDs (95% CI −3.39–−0.29). The combined mean reduction in antibiotic use across control counties was borderline significant, with an MBEC of −0.63 DDD/100BDs (95% CI −1.26–0.01). The reduction in the intervention county was not significantly greater than that in the control counties combined (Table 2).

Long-term NHs demonstrated the lowest usage of UTI-ABs, whereas short-term NHs exhibited the highest usage (Table 4). In the intervention county, a substantial reduction in UTI-ABs was observed, declining from 4.08 DDD/100BDs to 2.74 DDD/100BDs, with an MBEC of −1.34 (95% CI −1.85–−0.84) upon completion of the intervention (Table 2). In the control counties, there was a significant mean reduction in UTI-AB use, with an MBEC of −0.45 (95% CI −0.74–−0.15). However, when comparing the intervention and control counties collectively, the reduction in UTI-AB use was significantly greater in the intervention group (Table 4). The utilization of antibiotics within the intervention county for skin and soft-tissue infections (SSTI-ABs) decreased by 10.3%, while intravenous antibiotics (IV-ABs) decreased by 14.2%. Conversely, antibiotics for respiratory tract infections (RTI-ABs) increased by 8.6%, and other oral antibiotics increased by 7.7% within the intervention county. None of these changes exhibited statistical significance (Table 3). The changes observed in respiratory tract infection antibiotics (RTI-ABs), skin and soft-tissue infection antibiotics (SSTI-ABs), other oral antibiotics, and intravenous antibiotics (IV-ABs) within the control counties combined were not statistically significant. Additionally, no significant differences between the observed changes within the intervention and control counties were found when considering these antibiotic categories collectively.

No significant association was observed between the change in total antibiotic use in the intervention county and the proportion of staff who received educational presentations. This lack of association also extended to the use of the urinary dipstick checklist and the checklist for acute-onset disease in NHs within the intervention county. However, implementing the urinary dipstick checklist was correlated with a reduction in UTI-AB use in the intervention county, with a model-based difference in change of −0.95 (95% CI −1.86–−0.04). Conversely, there was no significant association between the proportion of staff receiving educational presentations or the use of the checklist for acute-onset disease and the change in UTI-AB use.

Our analysis did not reveal any statistically significant change in guideline-compliant antibiotic use within the intervention county or when combining the data from the control counties. Additionally, no significant difference was observed between the intervention and control counties when considering guideline compliance (Appendix A). Interestingly, larger NHs were associated with lower proportionate use of guideline-compliant antibiotics in the model (Appendix A).

### 2.3. Methenamine Use

Methenamine use decreased in the intervention county from 5.32 DDD/100BDs to 4.34 DDD/100BDs, with an MBEC of −1.34 (95% CI −2.24–−0.44) during the intervention period (Appendix A). The reduction was significantly greater than in the control counties (Appendix A).

## 3. Discussion

We observed a total reduction in antibiotic usage within the intervention NHs, although this reduction was not significantly higher than the reduction observed in the control counties combined. The lack of significant impact on total antibiotic use when compared to the reduction observed in the control counties combined raises questions concerning the effectiveness of the intervention strategy in achieving its primary outcome.

During the intervention year, the mean overall antibiotic use in the intervention county decreased by 15.8%. This reduction was most pronounced in short-term NHs, followed by mixed and long-term NHs. Notably, our findings align with a recent meta-analysis conducted by Wu et al. [30], which reported a mean 14% reduction in overall antibiotic use across 11 NH studies following the implementation of an antimicrobial stewardship strategy. While the control counties combined also experienced reductions in antibiotic use, the decrease was comparatively smaller than that observed in the intervention county. Among the control counties, we observed a notable variation in the extent of antibiotic use change. One county, Vestfold, exhibited a significant decrease, while others showed minimal to no change. The factors contributing to these differences were not explored in this study. However, a plausible explanation could be the increased attention given to AMR over the past decade, both within the scientific community and mainstream media, resulting in heightened awareness and local initiatives aimed at improving clinical practices. In the case of Vestfold (control 4), the county’s hospital initiated several antibiotic stewardship measures shortly before the study intervention. As patients often receive ongoing antibiotic treatment upon discharge from the hospital, these measures may have played a role in reducing antibiotic utilisation within the county’s primary care sector, including the NHs.

Notably, the use of oral antibiotics for UTIs decreased significantly in the intervention county compared to the control counties and constituted a large part of the total antibiotic reduction. This supports the notion that the intervention program may substantially impact antibiotic prescribing practices, particularly for UTIs, which are a common infection in NH residents. Our observed reduction in UTI-AB use of 33% corresponds well with a meta-analysis by Tandan et al. [27], who found a mean reduction of 37% in overall UTI antibiotic use (change per 1000 resident days) in seven NH antimicrobial stewardship intervention studies.

Given the high prevalence of UTIs and the inappropriate prescribing of antibiotics for ABU among NH residents [15,16,17,18,19,31,32], a major focus of the intervention was to promote the appropriate treatment of UTIs. In NHs, the prescription of UTI-ABs is often influenced by the observations reported by NH nurses and is predominantly carried out by NH physicians and out-of-hours physicians. Therefore, the intervention targeted both NH nurses and NH physicians, which likely played a significant role in the observed reduction of UTI-ABs. A qualitative study nested within our intervention highlighted several facilitators for this reduction, including enhanced ABU awareness, judicious use of point-of-care tests (including appropriate UTI diagnostic urine sampling and interpretation of results), the urinary dipstick checklist and heightened awareness of own antibiotic use [6].

The UTI checklist was well received by the NHs in the intervention county, as nearly all NHs regarded it as a valuable tool. Moreover, over two-thirds of the NHs anticipated that the checklist could reduce urine dipstick testing within their institutions. However, despite these positive perceptions, only approximately two-thirds of NHs reported actual implementation of the urine dipstick checklist at the midway point of the intervention. This discrepancy highlights the fact that a favourable perception alone is insufficient to achieve a high level of implementation. Successful implementation of quality improvement measures requires consideration of various factors, including education and training, the time aspect of the measure, personal and organisational motivation, culture, executive management, and effective communication and teamwork [33,34,35,36]. This observation highlights the multifaceted and interdependent nature of implementation challenges, influenced by both clinical and organisational complexities. It is plausible that a higher level of utilisation of the checklists could have been achieved if additional implementation resources were available during the intervention. Such increased utilisation may have resulted in more favourable outcomes than what was observed. Furthermore, our finding that the use of the urinary dipstick checklist was associated with a decrease in the use of UTI-ABs from baseline to intervention provides additional support for the notion that improving the implementation of the checklist could lead to better outcomes.

While the use of oral UTI-AB declined, usage of RTI-ABs and other oral antibiotics increased, although only slightly, from baseline to the conclusion of the intervention. Though not statistically significant, this trend raises concerns about potential compensatory effects, in which a decrease in one type of antibiotic category might be counterbalanced by an increase in another. It is important to note that the antibiotic indication groups used in our study were based on the findings from National Point Prevalence Surveys conducted by the Norwegian Institute of Public Health [4], and it is possible that these classification groups were somewhat loosely defined. Further investigation is warranted to better understand the underlying factors contributing to this potential shift in antibiotic prescribing practices.

Over the last decade, IV-ABs utilisation has seen an increasing trend in most Norwegian NHs following the implementation of the Norwegian Care Coordination Reform. The Government’s objective was to provide treatment to more NH residents on-site. However, the availability of IV administration also led to increased use of broad-spectrum agents outside of hospital settings. It is important to note that in Norway, no licensed oral formulations are available for second- or third-generation cephalosporins.

In our study, the proportion of IV-ABs in total antibiotic use at baseline was modest, accounting for approximately 14%. The intervention resulted in a non-significant reduction in the use of IV-ABs. One plausible explanation for this finding is that a substantial share of IV-ABs in NHs represents a continuation of treatment initiated during hospitalisation, which may be less likely to be questioned or reviewed by NH physicians [6].

The intervention strategy had more focus on appropriate diagnostics and use of antibiotics for UTIs compared to the attention given to other antibiotic classification groups, probably leading to the significant effect on UTI-AB and the lack of significant effects on the remaining antibiotic classification groups and on the total antibiotic use. As mentioned earlier, UTIs usually account for the lion’s share of infections and antibiotic use in NH residents and an assumed higher degree of unnecessary antibiotic use than other antibiotic classification groups. Accordingly, we therefore chose to intensify the focus on appropriate use of UTI-ABs, as we believed this approach would have a greater potential for reducing the antibiotic use, compared to the other antibiotic groups. An adjustment in the intervention strategy would be necessary to achieve the effects of this intervention for those other antibiotic classification groups.

Overuse of antibiotics is the leading driver of AMR [37], and broad-spectrum antibiotics contribute more than do narrower-spectrum agents. Moreover, use of broad-spectrum agents comes with an increased risk of unintended side effects such as, e.g., *Clostridioides difficile* infections, to which elderly frail residents are particularly vulnerable [38]. Unfortunately, although the educational part of the intervention included a lecture covering the national guidelines for antibiotic use in NHs, the intervention did not effectively influence the usage of guideline-compliant antibiotics. One reason may be a reluctance to change a hospital-initiated broad-spectrum prescription that the NH physician does not alter. Our intervention did not include, and thereby did not directly affect, hospital physicians, nurses and other health care workers. Previous studies have shown a lack of patient coordination between NHs and hospitals in Norway [6,39] which may lead to hospitalised patients receiving unnecessary broad-spectrum antibiotic treatment after discharge. Focusing more on improving this coordination in future interventions may lead to more guideline-compliant antibiotic use in NHs. Too high a level of broad-spectrum antibiotic treatment may also result from a lack of diagnostic and clinical data to differentiate between simple and complicated infections, a challenge in NHs that we have discussed elsewhere [4].

In Norway, methenamine hippurate is more commonly prescribed than in most other European countries, a trend which has been steadily increasing over the last three decades [1,11]. Although not an antibiotic, sensu stricto, and with no known resistance-driving capacity, the high use in Norway probably represents a certain degree of use without documented effect. Despite some recent studies showing promising effects of methenamine to prevent recurrent UTIs [40,41], the evidence base is weak, especially in the elderly, and more studies are needed [42]. We found a reduction of 20% in the use of methenamine hippurate in the intervention county during the intervention period, which was markedly higher than in the control counties. The use of methenamine was addressed at the intervention conferences, where the participants were urged to review long-lasting methenamine regimes at regular intervals, and to discontinue the treatment if there was doubt about the indication.

The intervention consisted of three well-known measures related to antibiotic stewardship programs: increasing professional knowledge through education, raising the awareness of one’s own antibiotic use, and supplying the institutions with tools to aid healthcare workers in the clinical assessment of possible infections. The intervention also indirectly involved residents and next-of-kin by disseminating information on ABU at the participating NHs. The observed variability in the proportion of NH staff exposed to the educational presentations raises concerns about the consistency and fidelity of the intervention across the participating NHs. Moreover, despite the Centre of Disease Control endorsing education as one of seven core elements in hospital antibiotic stewardship programs [43], our observed lack of association between educational presentations and antibiotic use in the intervention arm supports the finding that the measure alone is not sufficient to influence prescribing behaviours in NHs. However, the educational presentations may have proven more effective with more focus and resources. We carried out the project with limited personnel resources, which did not allow us to gather the participants for more than three large physical meetings during the intervention year, allowing us only to present the educational material to and train up to three resource persons from each NH to relay this material to their NH staff. Therefore, we infer that the quantity, and maybe even the quality, of educational efforts may have varied considerably across the NHs. The intervention strategy could also have been strengthened by better follow-up with the participating institutions, such as on-site visits to increase participant compliance with respect to the intervention measures. Improving the implementation strategy with increased external implementation support might have resulted in better results concerning the consistency and fidelity of the intervention, as well as a more effective intervention. On-site visits would have made it possible for the organizers to carry out part of the educational presentations at the NHs themselves, which could have increased the proportion of NH staff exposed to the educational material, increased the quality of the educational presentations and contributed to better training of the dedicated resource persons at the NHs. A high turnover rate of healthcare personnel in NHs is a known challenge, both in Norway and other countries [44,45], and may have further contributed to reduced adherence to the intervention at certain participating institutions. The low implementation rate of the acute-illness checklist can be attributed to the actuality that several institutions already had a well-implemented clinical scoring system (the Modified Early Warning Score) at the start of the intervention. These institutions felt that adding the acute illness checklist was unnecessary.

Based on our results and the findings in previous NH studies regarding coordination between NHs and hospitals [6,39], we believe that antibiotic use can be further improved by increasing the interaction and cooperation between the two healthcare levels. Discharge letters should ideally specify indication, duration and possibility of a switch from IV-ABs to oral antibiotics, if applicable. In addition, encouraging NH physicians and facilitating easy and accessible consulting with hospital colleagues will most likely contribute to increasing the prudent use of antibiotics for NH residents.

Our study had several strengths. It involved a large number of NHs spread across multiple counties, providing a broad geographic and demographic representation, and the real-world setting enhances the generalizability of our findings towards other countries with similar settings and healthcare systems similar to that of the Norwegian NHs. Furthermore, the diversity in type and size of the NHs participating in this study, in addition to our in-depth analysis of pharmacy sales data, offer a granular view of antibiotic usage. Adding to this, the use of routine data ensures complete follow-ups for all participating NHs. Finally, the intervention was multifaceted and involved a range of strategies, including education, feedback, and clinical guiding tools, all of which have been identified as practical and effective components of successful antibiotic stewardship programs.

However, our study also had several limitations which must be considered. Firstly, selection bias is a potential limitation when clustering individuals in studies and may represent a challenge even when the trials are randomised and controlled [46,47]. Our study design did not allow for randomisation of NHs associated with the intervention or control groups. Although the intervention county had a high participation rate and did not differ significantly from the control counties in terms of physician hours per bed per week and type of nursing home, we cannot rule out the possibility that there may have been unmeasured confounding factors that affected antibiotic use in the intervention and control counties. For example, physician factors, the degree of temporary healthcare worker use, the amount of nurse full-time equivalents, the residents’ ages and comorbidity compositions and the degree of involvement of relatives are possible confounding factors that could have affected the results of the intervention, but which we have not had the opportunity to control for in this study. Secondly, we relied on pharmacy sales data to measure antibiotic use, which may not reflect actual consumption at the patient level. Thirdly, we did not have data beyond the end of the 12-month intervention period, data which could have provided knowledge about the long-term effect of the intervention and antibiotic use patterns. Fourthly, we did not systematically evaluate the implementation outcomes, which makes it difficult to distinguish between the effectiveness of the implementation strategy and the effectiveness of the implementation. Finally, we did not evaluate the appropriateness of antibiotic prescribing, nor did we have access to data on complications in the participating NHs. Therefore, we cannot assess whether the observed reduction in antibiotic use found in our study carries any potential risks in terms of increased hospitalisations or mortality. However, several intervention studies of NHs showing significant antibiotic use reductions have not demonstrated any untoward consequences [26,28,48,49,50]. Thus, we do not consider it plausible that the demonstrated reduction of mainly oral UTI antibiotics could have significantly affected hospital admissions or mortality.

## 4. Materials and Methods

This section outlines the design and participants of the study, describes the intervention implemented, and details the methods of data collection and the categorisation of nursing homes and antibiotics.

### 4.1. Study Design and Participants

The current research is a multifaceted, controlled intervention study involving 33 nursing homes (NHs) situated in the county of Østfold, located in the southeastern region of Norway. Various types of NHs were invited to partake in the study, including those offering short-term care, long-term care, and both (“mixed NHs”). The recruitment strategy involved reaching out to the NHs through email and telephone. The intervention was endorsed at the level of the chief county and district medical officers in order to enhance participation. At the time of the invitation, none of the NHs were engaged in other antibiotic quality improvement initiatives. A questionnaire was circulated among the participating NHs at baseline, seeking general information about the number of wards, staff, and beds; type of ward; occupancy rate; and weekly physician hours. Four control counties (Telemark, Buskerud, Vestfold, and Agder) were selected based on their geographical proximity to Østfold. These counties did not have any parallel ongoing antibiotic quality-improvement programs and were not assigned to commence their interventions in the national RASK program until the conclusion of the intervention in Østfold. Baseline characteristics of the control counties were retrieved at enrolment in the national RASK program and controlled for by telephone contact up to the period examined in this study. The intervention was the first step in a nationwide intervention, with minor modifications in the program.

### 4.2. Intervention

The intervention, which spanned from October 2016 to October 2017, consisted of several key elements, as outlined in Figure 2.

### 4.3. Conferences and Antibiotic Feedback Reports

The intervention was initiated with a one-day conference, supplemented with two follow-up conferences at six-month intervals. For each participating NH, we developed antibiotic feedback reports that included the respondents’ own antibiotic use statistics, which were compared with those of NHs in the same category (long-/short-term and mixed NHs). NHs received the baseline antibiotic feedback reports one week prior to the start-up conference and the result reports immediately before the two follow-up conferences. These reports described current antibiotic use and changes compared with the baseline results.

### 4.4. Training, Clinical Decision Support Tools and Academic Detailing

The start-up conference included several educational presentations focused on the accurate diagnostics, treatment, and prevention of infections, with particular emphasis on UTIs and ABU. Additional topics included the national guidelines for antibiotic use in NHs, how to communicate antibiotic statistics within their respective NH, and the ethical aspects of antibiotic use. Clinical checklists were introduced, targeting nursing staff, concerning the correct indication for urinary dipstick testing and early recognition of acute-onset disease. The urinary dipstick checklist detailed specific and non-specific signs and symptoms of UTIs, aiming to raise the threshold for unnecessary urinary dipstick testing. Instructions were provided during the start-up conference on how to incorporate these into daily clinical practice. NHs were recommended to ensure that physicians did not assess a urine dipstick result unless accompanied by a completed checklist. The checklist for early recognition of acute-onset disease was designed as a tool for auxiliary nurses to identify possible cases of new conditions or acute deterioration in NH residents, which should then be reported to registered nurses for further investigation.

Workshops, plenary discussions, and an academic detailing session were also organised, in which participants discussed their antibiotic use in groups under the supervision of organisers from the Antibiotic Centre for Primary Care. The antibiotic reports were actively used during these academic detailing sessions.

### 4.5. Resource Persons

Each nursing home had up to three representatives participating in the conferences, typically consisting of a physician, one or several nurses, and occasionally a person with managerial responsibilities at the NH. At the start-up conference, each NH designated one or more of the attending participants as their responsible contact representatives for the intervention year.

### 4.6. Train-the-Trainer Model, Educational Material and Written Resources

We used a train-the-trainer approach by first educating the NH representatives and the dedicated resource persons in the topics in focus, and then providing guidance on how the resource persons should train the NH staff, including sharing the educational material. The resource persons were responsible for the education in their respective nursing homes and for implementing the clinical decision support checklists into daily clinical practice. All educational materials and the two clinical checklists presented at the start-up conference were made available online and by email to the participating NHs. An information brochure on asymptomatic bacteriuria, which was aimed at residents, their next-of-kin and healthcare workers at the institutions, was developed for the intervention and shared with the participating NHs online and by email.

### 4.7. Data Collection, Antibiotic Classification, and Nursing Home Characteristics

The resource persons were required to provide a written report detailing the implemented measures within six months of the intervention. The reports were distributed to and retrieved from the participating NHs by email, and the responses were converted to a separate file for statistical analyses by the organisers of the intervention. Baseline antibiotic sales data covering 12 months (October 2015 to October 2016) were extracted electronically from the pharmacies’ central databases for all participating NHs prior to the intervention. The contents of the antibiotic sales data files were converted to separate files for statistical analyses by the organisers of the intervention. The data included antibiotics for systemic use (ATC group J01), oral metronidazole (P01AB01), oral rifampicin (J04AB02) and oral vancomycin (A07AA09). In addition, the extracted data contained dates of purchase, customer characteristics, generic drug name, Anatomical Therapeutic Chemical (ATC) code, administration form and total DDDs from all participating NHs. The intervention period data (October 2016 to October 2017) were extracted at the midpoint (six months) and the end (12 months) of the intervention.

The Anatomical Therapeutic Chemical Index (ATC/DDD) version 2019 was used [51]. Antibiotics were grouped into five categories: (1) oral urinary-tract infection antibiotics (UTI-ABs), (2) oral respiratory tract infection antibiotics (RTI-ABs), (3) oral skin-and-soft-tissue infection antibiotics (SSTI-ABs), (4) all other oral antibiotics, and (5) intravenous antibiotics (IV-ABs). Methenamine (J01XX05), a urinary tract antiseptic rather than an antibiotic, was classified and analysed separately from the other groups.

The NHs were classified based on their residency into long-term, short-term, or mixed (long-term and short-term) NHs. In addition, NHs were categorised by their bed count into small NHs (0–40 beds), medium NHs (41–69 beds), and large NHs (≥70 beds). Additional details regarding justification of the antibiotic classification groups and additional information on NH characteristics have been published elsewhere [4].

### 4.8. Statistical Analyses

Antibiotic use was measured in DDD/100 BD for each individual NH. A total weighted mean was then calculated for all NHs within each NH category and per county, taking into account the total bed capacity. These calculations were conducted for both the baseline and intervention years. To assess alterations in antibiotic consumption pre- and post-intervention, linear mixed models were employed at the institution level. These models were employed to estimate and test for changes in antibiotic usage from baseline to after-intervention within and between the intervention and control counties. The main model included DDD/100 BD as the outcome variable. It also integrated time (before–after intervention), an intervention indicator (intervention–control) and the interaction between time and the intervention indicator. Additionally, size of NH (categorised), NH resident category and doctor hours/bed/week were included as additional independent variables. A random intercept at the NH level was introduced to handle dependence in the data. Furthermore, at the institutional level, linear mixed models were utilised to estimate and test for potential associations between certain implementation measures undertaken and alterations in overall and UTI-AB use from baseline to intervention within the intervention county. In lieu of the intervention indicator used in the previous model, an exposure variable was incorporated for each executed model. This exposure variable encompassed specific implementation measures, including the proportion of staff exposed to educational presentations, the utilisation of the urinary dipstick checklist (use/no use) and the application of the checklist for acute-onset diseases (use/no use). Similar to the prior model, the updated model included DDD/100 BD as the outcome variable, time (before–after intervention) and the interaction between time and the exposure variable. Independent variables encompassed NH size (categorised), NH resident category and doctor hours/bed/week. A random intercept at the NH level was included to handle data interdependence. In all regression analyses, we applied a probability-weighted model. This approach was based on each NH’s portion of the total bed count for the intervention and combined control counties (probability weight = number of beds per institution/number of beds for all counties combined). By applying this probability-weighted framework to the regression analyses, changes in antibiotic usage within larger NHs exerted a more pronounced impact on the overall outcome compared to similar variations within smaller NHs, and vice versa. We used STATA/SE^®^ 17.0 statistics programs for all statistical analysis [52].

## 5. Conclusions

Our study implies that multifaceted interventions based on principles from antibiotic stewardship programs with academic detailing can be successfully disseminated in Norwegian NHs. With limited resources, the intervention strategy did not prove successful in reducing the total antibiotic use but had a significant impact in reducing the antibiotic use for urinary tract infections in NHs. The observed changes highlight the complexities inherent in modifying clinical practices across diverse institutions, and further research is needed to explore the long-term impacts of such interventions, in terms of both antibiotic use and clinical outcomes for NH residents. Future interventions should consider bolstering inter-institutional coordination, especially between NHs and hospitals, and ensuring consistent and high-fidelity implementation to further optimize antibiotic use in NHs.

## Figures and Tables

**Figure 1 antibiotics-12-01372-f001:**
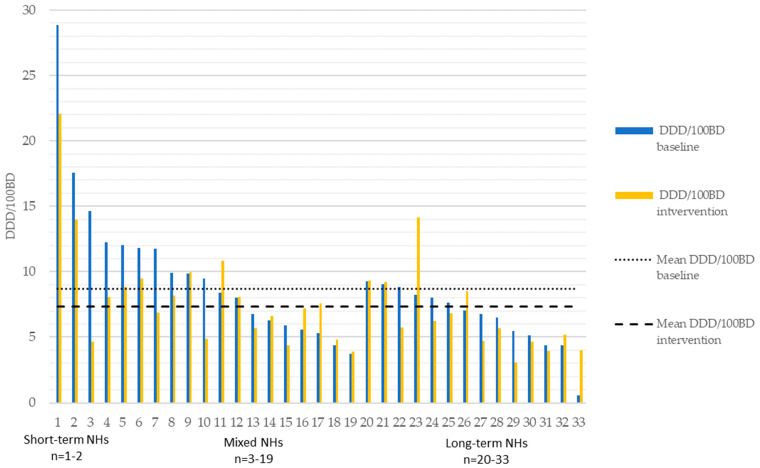
Changes in total antibiotic use in DDD/100 BD per institution pre- and post-intervention-period for the intervention county, October 2015–October 2016 versus October 2016–October 2017.

**Figure 2 antibiotics-12-01372-f002:**
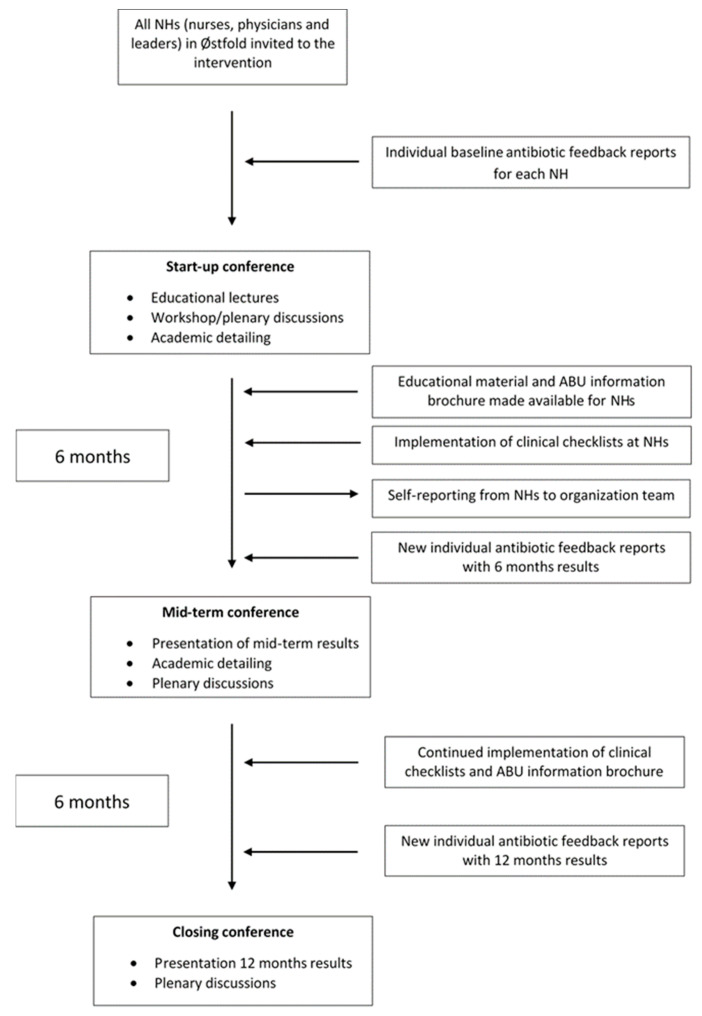
Flow chart with an overview of the main elements and measures in the 12-month “RASK” intervention in Østfold County.

**Table 1 antibiotics-12-01372-t001:** Baseline characteristics of intervention and control counties.

	Intervention	Control 1	Control 2	Control 3	Control 4	Controls (Total)
County population 2016	289,267	277,684	298,487	172,494	244,967	993,632
Number of NHs	33	35	46	25	30	136
Number of beds	1807	1477	1845	1090	1561	5973
						Controls (mean)
Doctor hours/bed/week	0.46	0.54	0.43	0.6	0.45	0.49
Beds/NH	55 (r: 13–120)	42 (r: 13–120)	40 (r: 8–91)	44 (r: 16–100)	52 (r: 13–126)	44 (r: 8–126)
Bed days/NH/year	19,840	14,802	14,177	14,967	18,695	15,480
Size of NHs						
0–40 beds	9	19	27	13	11	17.5
41–69 beds	15	14	12	9	11	10.75
70+ beds	9	2	7	3	8	5
NH category						
Short-term NHs	2	3	1	2	3	2.75
Long term NHs	14	8	17	6	12	10.75
Mixed NHs	17	24	28	17	15	21

**Table 2 antibiotics-12-01372-t002:** Linear mixed model to identify the model-based estimated comparison of changes in total antibiotic use, pre- (October 2015–October 2016) versus post intervention (October 2016–October 2017) between the intervention county and control counties combined (number of observations: 338).

	Estimated Difference β	95% CI	*p*
Change in total antibiotic use in control counties combined	ref.			
Comparison of change in total antibiotic use between intervention and controls combined	−0.75	−1.91–0.41	0.207
Nursing home category:			
Mixed Long-term Short-term	ref. −1.56 12.45	−2.51–−0.62 6.86–17.96	0.001 <0.001
Size of nursing home:			
Small	ref.		
Medium	−1.83	−3.22–−0.43	0.01
Large	−2.33	−4.15–−0.52	0.012
Doctor hours/bed/week	0.03	−0.01–0.07	0.112

**Table 3 antibiotics-12-01372-t003:** Total mean antibiotic use per indication group in DDD/100 BD, and the model-based * estimated changes identified ** pre- (October 2015–October 2016) and post-intervention (October 2016–October 2017) for the intervention county and the control counties combined.

Antibiotic Indication Group	County	DDD/100 BD Baseline	DDD/100 BD Intervention	Estimated Change β	95% CI	*p*
Oral UTI	Intervention	4.08	2.74	−1.34	−1.85–−0.84	<0.001
	Control	3.69	3.25	−0.45	−0.74–−0.15	0.003
Oral RTI	Intervention	2.29	2.49	0.2	−0.17–0.56	0.29
	Control	3.15	3.17	0.01	−0.3–0.31	0.97
Oral SSTI	Intervention	0.70	0.63	−0.07	−0.33–0.2	0.62
	Control	0.61	0.51	−0.1	−0.2–0.01	0.08
Other oral antibiotics	Intervention	0.36	0.38	0.03	−0.11–0.18	0.65
	Control	0.32	0.27	−0.05	−0.15–0.05	0.34
Intravenous	Intervention	1.20	1.03	−0.16	−0.37–0.05	0.15
	Control	0.76	0.72	−0.04	−0.2–0.11	0.59
Total	Intervention	8.68	7.31	−1.38	−2.35–−0.41	0.005
	Control	8.53	7.93	−0.63	−1.27–0.01	0.054

* Linear mixed model regression analysis. ** All models adjusted for NH size, NH category and doctor hours/bed/week.

**Table 4 antibiotics-12-01372-t004:** Linear mixed model to identify the model-based comparison of changes in UTI antibiotic use, pre- (October 2015–October 2016) versus post intervention (October 2016–October 2017) between the intervention county and the control counties combined (number of observations: 336).

	Estimated Difference β	95% CI	*p*
Change in UTI-AB use in combined control counties	ref.		
Comparison between changes in UTI-AB use within intervention and combined controls	−0.9	−1.28–−0.31	0.003
Nursing home category:			
Mixed	ref.		
Long-term	−0.83	−1.3–−0.36	0.001
Short-term	5.43	2.8–8.06	<0.001
Size of nursing home:			
Small	ref.		
Medium	−0.61	−1.27–0.06	0.073
Large	−0.75	−1.58–0.09	0.081
Doctor hours/bed/week	0.005	−0.01–0.02	0.57

## Data Availability

Extended data are available upon request to the corresponding author (N.J.H.).

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
