# Peer review of "A Multifaceted Intervention and Its Effects on Antibiotic Usage in Norwegian Nursing Homes"

_antibiotics, 2023, doi:10.3390/antibiotics12091372_

Round 1
Reviewer 1 Report
The manuscript entitled “ A multifaceted intervention and its effects on antibiotic usage 2 in Norwegian nursing homes” is presenting a very important issue about antibiotic use. It has educational value in tracking and dealing with antibiotic administration in NHs.
In my final opinion, the work is very well written and planned. Authors are aware of the limitations of their work and wrote about it. I have only some minor suggestions:
-Line 94: Please use „the aim of the study was to present..”
-Line 224: UTI was already explained (write full name or abbreviation)
-Line 225: ABU was already explain – do not repeat
-avoid first-person narration, Authors should use mainly passive voice (the exception from that is the discussion section).
Author Response
Response to Reviewer 1 Comments :
Point 1: The manuscript entitled “A multifaceted intervention and its effects on antibiotic usage in Norwegian nursing homes” is presenting a very important issue about antibiotic use. It has educational value in tracking and dealing with antibiotic administration in NHs.
In my final opinion, the work is very well written and planned. Authors are aware of the limitations of their work and wrote about it
Response 1: Thank you for reviewing and commenting on our manuscript and that you find the paper valuable in terms of tracking and dealing with antibiotic usage in nursing homes.
Point 2: Line 94: Please use „the aim of the study was to present..”
Response 2: We have revised the manuscript accordingly to the reviewers comment (line 94).
Point 3: Line 224: UTI was already explained (write full name or abbreviation)
Response 3: The manuscript have been revised accordingly (line 228).
Point 4: Line 225: ABU was already explain – do not repeat.
Response 4: The manuscript have been revised accordingly (line 229).
Point 5: Avoid first-person narration, Authors should use mainly passive voice (the exception from that is the discussion section).
Response 5: We have actively revised the introduction, results and methods section with a passive voice approach.
Reviewer 2 Report
Dear prof
please can I know if there are gender differences in this antibiotics use?
however can I known if this prescription is related to clinicians or not?
thank you
Dear prof
please can I know if there are gender differences in this antibiotics use?
however can I known if this prescription is related to clinicians or not?
thank you
Author Response
Response to Reviewer 2 Comments:
Point 1: Please can I know if there are gender differences in this antibiotics use?
Response 1: Thank you for taking your time to evaluate and comment on our manuscript Unfortunately, our study is based solely on pharmacy sales data, not actual antibiotic prescriptions, leaving us without results/data on which residents have received antibiotic treatment/prescriptions during the baseline and intervention periods. Therefore, it is not possible to present data on gender differences in the current study.
Point 2: however can I known if this prescription is related to clinicians or not?
Response 2: We agree that a number of various factors related to prescribers would have added valuable information for our research and future planning of antibiotic quality improvement programs in nursing homes. Unfortunately, we do not have sufficient adequate data at the clinician level to investigate possible confounding factors. The only information related to clinicians available is the doctor hours/bed/week per nursing home, which we have adjusted for in the mixed linear regression models applied in the study. We did not find any correlation related to the amount of doctor hours/bed/week and antibiotic use in the study.
Reviewer 3 Report
Overall, while the study suggests some positive impact on reducing UTI-AB use, there are several limitations and inconsistencies in the findings and methodology, which call for further research and replication to draw more robust conclusions about the effectiveness of the intervention on antibiotic usage in nursing homes. The specific comments are below:
1:Lack of a significant impact on total antibiotic use: The study claims that the multifaceted intervention led to a 15.8% reduction in total antibiotic use, but this reduction was not significantly greater than the reduction observed in the control counties. This raises questions about the effectiveness of the intervention in achieving its intended outcome.
2: Limited impact on specific antibiotic categories: While the study reports a significant reduction in oral antibiotic usage for urinary tract infections (UTI-AB), the reductions in other antibiotic categories such as respiratory tract infections (RTI-ABs), skin and soft tissue infections (SSTI-ABs), and intravenous antibiotics (IV-ABs) were not statistically significant. This suggests that the intervention may have had a limited effect on a broader range of antibiotic usage patterns.
3: Inconsistent implementation of intervention material: The study mentions that the proportion of staff members receiving educational presentations varied substantially, ranging from 15% to 95%. Such a wide variation in implementation raises concerns about the consistency and fidelity of the intervention across the participating nursing homes, which could have influenced the overall impact of the intervention.
4: Lack of association between educational presentations and antibiotic use: The analysis did not find a significant association between the proportion of staff receiving educational presentations and the change in total antibiotic use or guideline-compliant antibiotic use. This raises questions about the effectiveness of the educational sessions in influencing prescribing behaviors among healthcare providers.
5: Inadequate assessment of guideline compliance: The study did not observe any statistically significant change in guideline-compliant antibiotic use within the intervention county or when comparing it to the control counties. This suggests that the intervention may not have effectively influenced adherence to established guidelines for antibiotic prescribing.
6: Incomplete data reporting: The study does not provide detailed information on the methodology used to collect and analyze the data. Important aspects such as data collection procedures, statistical methods, and potential confounding factors are not adequately described, which limits the transparency and replicability of the study.
7: Lack of long-term follow-up: The study only assessed the impact of the intervention over a 12-month period. Antibiotic usage patterns and the sustainability of the observed changes in the long term are important considerations that were not addressed. Without long-term follow-up data, it is challenging to assess the lasting effects of the intervention.
8: Lack of generalizability: The study was conducted in a specific region of Norway, which limits the generalizability of the findings to other nursing home settings or healthcare systems. The unique characteristics of the study population and the specific context in which the intervention was implemented may limit its applicability to other settings.
Minor editing of English language required
Author Response
Response to Reviewer 3 Comments:
Point 1: Overall, while the study suggests some positive impact on reducing UTI-AB use, there are several limitations and inconsistencies in the findings and methodology, which call for further research and replication to draw more robust conclusions about the effectiveness of the intervention on antibiotic usage in nursing homes.
Response 1: We would like to thank Reviewer 3 for taking the time to evaluate our manuscript with several valuable questions, concerns and suggestions for consideration.
Point 2: Lack of a significant impact on total antibiotic use: The study claims that the multifaceted intervention led to a 15.8% reduction in total antibiotic use, but this reduction was not significantly greater than the reduction observed in the control counties. This raises questions about the effectiveness of the intervention in achieving its intended outcome.
Response 2: The reviewer is correct in pointing out that one of the main aims of the intervention was the change in total antibiotic usage compared to the change in the control counties combined. Unfortunately, we cannot conclude from our data that the intervention was effective regarding this intended outcome. We agree that this has not been addressed clearly enough in the manuscript and have therefore revised the manuscript accordingly in the abstract section (lines 26-28), the discussion section (lines 198-200) and the conclusions section (lines 525-532).
Point 3: Limited impact on specific antibiotic categories: While the study reports a significant reduction in oral antibiotic usage for urinary tract infections (UTI-AB), the reductions in other antibiotic categories such as respiratory tract infections (RTI-ABs), skin and soft tissue infections (SSTI-ABs), and intravenous antibiotics (IV-ABs) were not statistically significant. This suggests that the intervention may have had a limited effect on a broader range of antibiotic usage patterns.
Response 3: The reviewer correctly points out that the intervention had a limited effect on the other antibiotic classification groups apart from UTI-ABs. In the manuscript, we have discussed possible causes for this lack of effect (e.g. our suggestion of a possible compensatory effect, reluctance to change hospital initiated IV-AB treatments (lines 256-276)), but agree that this section should be addressed more thoroughly. We have, therefore, added a section regarding the focus and intensity of the intervention towards UTI-ABs; and that less focus on other antibiotic classification groups has led to less reduction in these (lines 277-287).
Point 4: Inconsistent implementation of intervention material: The study mentions that the proportion of staff members receiving educational presentations varied substantially, ranging from 15% to 95%. Such a wide variation in implementation raises concerns about the consistency and fidelity of the intervention across the participating nursing homes, which could have influenced the overall impact of the intervention.
Response 4: We agree that the wide variation in reported distribution of the intervention material raises concerns about the consistency and fidelity of the intervention and concur that this should be been pointed out more thoroughly in the manuscript. We have therefore revised the manuscript in an attempt to adequately address this in the discussion section (lines 320-341 and lines 380 – 383). Moreover, we think the added text also addresses the reviewer’s comments in “Point 5” regarding the concerns towards effectiveness of educational sessions in influencing prescribing behaviours. We have also added an extra citation related to this topic as a reference #43: https://www.cdc.gov/antibiotic-use/core-elements/hospital.html
Point 5: Lack of association between educational presentations and antibiotic use: The analysis did not find a significant association between the proportion of staff receiving educational presentations and the change in total antibiotic use or guideline-compliant antibiotic use. This raises questions about the effectiveness of the educational sessions in influencing prescribing behaviors among healthcare providers.
Response 5: Please see our answer in “Response 4”.
Point 6: Inadequate assessment of guideline compliance: The study did not observe any statistically significant change in guideline-compliant antibiotic use within the intervention county or when comparing it to the control counties. This suggests that the intervention may not have effectively influenced adherence to established guidelines for antibiotic prescribing.
Response 6: The results of the study clearly illustrates that the intervention did not effectively influence an increased adherence to the national guidelines in terms of choice of antibiotics during the intervention period. Although already partly addressed and discussed in the discussion section of the manuscript, we agree that a broader discussion on this important topic is warranted. We have therefore revised the part regarding hospital initiated broad spectrum antibiotics and added an assessment regarding the interventions focus on reducing inappropriate UTI-AB treatments (lines 291 – 303).
Point 7: Incomplete data reporting: The study does not provide detailed information on the methodology used to collect and analyse the data. Important aspects such as data collection procedures, statistical methods, and potential confounding factors are not adequately described, which limits the transparency and replicability of the study.
Response 7:
- Regarding data collection, we have added a more detailed description on the procedures concerning how the data was retrieved in the materials and methods section (lines 465 – 475). If the reviewer still finds that important aspects concerning data collection is lacking, we would kindly ask the reviewer to be more specific as to what is warranted.
- Regarding more detailed information concerning the statistical analysis, we have added additional information concerning this aspect in the materials and methods section (lines 490-520). Again, we would kindly ask the reviewer to be more specific if additional information is judged pertinent.
- Regarding the issue of potential confounding factors not being sufficiently described, we would point out that this has been discussed in lines 368 – 377 in the discussion (limitations) section. We have added another possible confounding factor to the discussion (line 373). If the reviewer still finds the discussion insufficient on the topic of confounders, we would be grateful for some suggestions.
Point 8: Lack of long-term follow-up: The study only assessed the impact of the intervention over a 12-month period. Antibiotic usage patterns and the sustainability of the observed changes in the long term are important considerations that were not addressed. Without long-term follow-up data, it is challenging to assess the lasting effects of the intervention.
Response 8: We certainly agree with the referee that this limitation is important and should be addressed under “limitations” in the discussion section of the manuscript. We have revised the manuscript accordingly (lines 378-380).
Point 9: Lack of generalizability: The study was conducted in a specific region of Norway, which limits the generalizability of the findings to other nursing home settings or healthcare systems. The unique characteristics of the study population and the specific context in which the intervention was implemented may limit its applicability to other settings.
Response 9: The reviewer is correct in pointing out that the study may have a lack of generalizability towards nursing homes in other settings or belonging to other healthcare systems. We still believe that the findings of the study reflect a high degree of generalizability for nursing homes in other parts of Norway, as the NH setting across Norway is relatively uniform and the same healthcare system encompass all parts of the country. We also believe that the findings reflect a certain degree of generalizability towards NHs in other countries, given that they inhabit a similar NH system as Norway. Based on the reviewer’s comments regarding this potential limitation we have added a clarification in the discussion (lines 357– 358).
Round 2
Reviewer 2 Report
it is ok for me
ok
Author Response
We are grateful for the time the reviewer have taken to evaluate and comment on our manuscript.
Reviewer 3 Report
I do not have any further comments.
Author Response
We would like to thank the reviewer for the time and effort spent in evaluating and commenting our paper.